# Multi Locus Sequence Typing and *spa* Typing of *Staphylococcus aureus* Isolated from the Milk of Cows with Subclinical Mastitis in Croatia

**DOI:** 10.3390/microorganisms9040725

**Published:** 2021-03-31

**Authors:** Luka Cvetnić, Marko Samardžija, Sanja Duvnjak, Boris Habrun, Marija Cvetnić, Vesna Jaki Tkalec, Dražen Đuričić, Miroslav Benić

**Affiliations:** 1Department of Bacteriology and Parasitology, Croatian Veterinary Institute, 10000 Zagreb, Croatia; lcvetnic@veinst.hr (L.C.); marjanovic@veinst.hr (S.D.); habrun@veinst.hr (B.H.); benic@veinst.hr (M.B.); 2Clinic for Obstetrics and Reproduction, Faculty of Veterinary Medicine University of Zagreb, 10000 Zagreb, Croatia; 3Department of Microbiology and Infectious Diseases, with Clinic, Faculty of Veterinary Medicine University of Zagreb, 10000 Zagreb, Croatia; mcvetnic@vef.hr; 4Veterinary Laboratory Križevci, Croatian Veterinary Institute, 10000 Zagreb, Croatia; jaki.vzk@veinst.hr; 5Mount Trade Company, 43280 Garešnica, Croatia; dduricic19@gmail.com

**Keywords:** *S. aureus*, mastitis, MLST, *spa* typing

## Abstract

Background: The bacterial species *S. aureus* is the most common causative agent of mastitis in cows in most countries with a dairy industry. The prevalence of infection caused by *S. aureus* ranges from 2% to more than 50%, and it causes 10–12% of all cases of clinical mastitis. Aim: The objective was to analyze 237 strains of *S. aureus* isolated from the milk of cows with subclinical mastitis regarding the *spa, mecA*, *mecC* and *pvl* genes and to perform *spa* and multi-locus sequence typing (MLST). Methods: Sequencing amplified gene sequences was conducted at Macrogen Europe. Ridom StaphType and BioNumerics software was used to analyze obtained sequences of *spa* and seven housekeeping genes. Results: The *spa* fragment was present in 204 (86.1%) of strains, while *mecA* and *mecC* gene were detected in 10 strains, and the *pvl* gene was not detected. *Spa* typing successfully analyzed 153 tested isolates (64.3%), confirming 53 *spa* types, four of which were new types. The most frequent *spa* type was t2678 (14%). MLST typed 198 (83.5%) tested strains and defined 32 different allele profiles, of which three were new. The most frequent allele profile was ST133 (20.7%). Six groups (G) and 15 singletons were defined. Conclusion: Taking the number of confirmed *spa* types and sequence types (STs) into account, it can be concluded that the strains of *S. aureus* isolated from the milk of cows with subclinical mastitis form a heterogenous group. To check the possible zoonotic potential of isolates it would be necessary to test the persons and other livestock on the farms.

## 1. Introduction

The bacterial species *S. aureus* is the most common causative agent of mastitis in cows in most countries with a dairy industry. The prevalence of infection caused by this bacterium ranges from 2% to more than 50%, and it causes 10–12% of all cases of clinical mastitis [1,2,3]. Cvetnić et al. [4] investigated quarter udder milk samples from 385 cows in Croatian dairy farms and found 13.1% infected udder quarters. At least one udder quarter was infected in 29.2% of cows examined. *S. aureus* was isolated from 4.48% of udder quarter samples.

Methicillin-resistant *Staphylococcus* was first confirmed as the causative agent in cow mastitis in 1972 [5]. The spread of infection throughout the herd is possible through the broad contact between cows via milking machines and equipment [6,7,8,9].

The polymerase chain reaction (PCR) multiplex tests for a combination of different genes for the rapid identification of strains of *S. aureus*.

For more detailed identification of *S. aureus*, molecular typing techniques are applied. 

The multi-locus sequence typing (MLST) of *S. aureus* is based on the analysis of seven *housekeeping* genes (*arcC*, *aroE*, *glpF*, *gmk*, *pta*, *tpi* and *ygiL*) according to Enright et al. [10]. Order of nucleotides within genes enables to differentiate strains, while the combination of alleles make the allele profile or sequence type (ST) [11].

*Spa* typing to investigate a single gene locus is often used in combination with MLST. It supports the MLST defined clonal complexes relatively accurately, with some exceptions. However, the results have limited value in phylogenetic studies [12].

Population genetic studies have identified the genotypes of *S. aureus* associated with specific host species. The individual sequence types (ST) ST71, ST97, ST126, ST133 and ST151 have been confirmed among ruminants [13,14], ST5 in poultry [15] and ST433 in pigs [16]. The dominant types in humans are ST1, ST5, ST8, ST22, ST30, ST36, ST45, ST57, ST80, ST228, ST239, ST247 and ST250 [17]. It would appear that specific sequence types, such as ST22, ST254 and ST398 have a broad spectrum of hosts, and can adapt to both animals and humans [18,19]. Sheet et al. [20] showed the affiliation of clone complexes CC5, CC97, CC133 and CC151 in 70 isolates of *S. aureus* from cow milk samples in Germany. The predominant genotype was CC133, which accounted for 27.1% of the total. All isolates were methicillin-sensitive (MSSA). In Denmark, cow milk samples contained ST398, which colonizes cattle, other animal species, and humans [21].

In Algeria, a wide diversity of *spa* types was established, though the clone complexes CC97 and CC22 were dominant in cows, and the typical animal clone CC97 was also occasionally detected in humans [22]. Holmes and Zadoks [23] reported the isolation of CC130, ST425 and CC1943 from humans, even though these types were specific for cattle. In a Belgian study of livestock-associated methicillin resistant *S. aureus* (LA-MRSA) in the noses of dairy cows, carriers were detected in 19.8% of all samples, and most isolates were ST398, while some isolates belonged to the types ST239 and ST8. ST8 strains are typically of human origin, although these have recently also been detected in cattle [24].

Schmidt et al. [25] examined staphylococci isolated from the milk of cows with mastitis and in nasal swabs of humans in close contact with cows. Seven sequence types were detected: ST8 (CC8), ST97 (CC97), ST351 (CC705), ST352 (CC97), ST508 (CC45), ST2992 (CC97) and ST3538 (CC97). The cattle and human isolates were identical at three sampling sites, and could not be differentiated (ST97, ST352, CC97 and ST8).

A review of the comparable literature shows the corroboration between strains of the species *S. aureus* isolated from humans with those isolated from animals has been confirmed multiple times. Most of these studies have applied molecular techniques, or only MRSA strains were selected in the research. Therefore, we also included MSSA in this study.

The aims of the study were to: Confirm the presence of the *spa* gene, methicillin-resistance genes (*mecA* and *mecC*) and the *pvl* gene, perform sequence analysis of the *spa* gene to confirm its polymorphism, and to perform MLST analysis to define the sequence types isolated from mastitis cases in cows. 

## 2. Methods

Strains of *S. aureus* used in the research were picked up from the collection gathered during the six years (from 2014 to 2019) and stored in trypticase soy broth (TSB) (Merck, Darmstadt, Germany) with 30% sterile glycerin at −20 °C. All *S. aureus* strains were isolated from individual quarter udder milk samples of cows suffering from subclinical mastitis. Cows were considered as having subclinical mastitis based on positive California mastitis test (CMT) and pathogen isolation without clinical signs of mastitis. The samples were taken by field veterinarians or by the laboratory personnel during routine visits to farms, sent to the laboratory and cultivated on aesculin blood agar. Suspected colonies were Gram stained, checked for catalase production and subcultivated on Baird–Parker agar and blood agar agar [26]. Furthermore, the tube coagulase-test was carried out as well as the identification by semi-automated identification system Micronaut RPO (Merlin Diagnostika GmbH, Berlin, Germany). After thawing, the isolates were revitalised on aesculin blood agar. For further molecular probes we chose one or two typical isolates per herd avoiding double selection. For this study, we selected a total of 237 strains identified as *S. aureus* originating from 162 herds in 16 counties and isolated in the specified period (Table 1). We confirm that all methods were carried out in accordance with relevant guidelines and regulations and that all experimental protocols were approved by an Ethical Committee of the Faculty of Veterinary Faculty University of Zagreb, Croatia. 

### 2.1. DNA Isolation

In order to confirm the species and to assess the resistance toward methicillin, revitalized *S. aureus* culture was suspended in 100 µL of sterile DNA/RNA free distilled water (MilliQ water, in-house production) in 2 mL test tubes. This mixture was heated for 20 min at 95 °C in a thermoblock with constant stirring. Then the test tubes were centrifuged at 14,000× *g* for 1 min. For the test we used 2 or 5 µL of supernatant.

In order to obtain DNA from strains for further genotyping the commercially available kit QIAcube DNA Mini Kit (QIAGEN, Hilden, Germany) according to the manufacturer’s instructions. For the test, we used 2 or 5 µL of supernatant. Bacterial culture was stirred into 180 µL ATL buffer, then 20 µL proteinase K was added, and the mixture was incubated at 56 °C until complete lysis. These prepared samples were then processed with the QIAcube instrument (QIAGEN, Hilden, Germany) to isolate DNA. Further steps prior to processing were: 200 µL AL buffer was added to the obtained solution, and incubated at 70 °C for 10 min; then 200 µL ethanol (96–100%) was added and the mixture briefly centrifuged. The contents of the test tube were transferred to a filtering test tube and centrifuged at 6000× *g* for 1 min. Then 500 µL AW1 buffer was added and centrifuged at 6000× *g* for 1 min. The procedure was repeated with AW2 buffer and centrifuged at full speed for 3 min. The contents of the filter test tube were transferred to a clean 1.5 mL test tube, and 200 μL AR buffer was added, incubated for 1 min at room temperature, and centrifuged at a speed of 6000× *g* for 1 min. The final step with AE buffer was repeated to obtain a higher quantity of isolated DNA.

### 2.2. Multiplex Polymerase Chain Reaction (PCR)

This method is based on the amplification of the four genes that prove the presence of the species *S. aureus* (*spa* gene), proving methicillin resistance by the presence of the genes *mecA* and *mecC* and the gene for *pvl*. The primers shown in Table 2 were used in the PCR [27].

*Spa* fragment sizes ranged from 180–600 bp depending on the *spa* type, the *mecA* fragment is 162 bp, and the *mecC* 138 bp in size. Amplified 85 bp fragments indicate the presence of the gene that codes for Panton–Valentine leukocidin (*pvl*). *S. aureus* ATCC 700699 was used as positive control. 

### 2.3. Spa Typing

The fragments obtained by amplification of the *spa* region by the polymerase chain reaction (PCR) were sequenced and assigned by numerical code that is based on the order of nucleotides. We amplified the specific repeating region of the gene responsible for the synthesis of protein A (*Spa*) [28]. The primer listed in Table 2 were used for the PCR.

### 2.4. Multi-Locus Sequence Typing (MLST)

MLST is a method based on determining the sequences of DNA fragments about 450 bp in length in the “housekeeping” genes of *S. aureus* (*arcC*, *aroE*, *glpF*, *gmk*, *pta*, *tpi* and *ygiL*) [10]. The obtained sequence type, i.e., allele profile, corresponds to the sequence type (ST). On the basis of obtained ST types, the clones are grouped into clonal complexes (CC) using the eBURST (Based Upon Related Sequence Types) software package (http://eburst.mlst.net/; accessed on 8 March 2019) and the database available on the MLST website (http://www.mlst.net/; accessed on 8 March 2019) [11]. Strains of *S. aureus* are grouped into the same CC when they have identical sequences in five of the seven housekeeping genes, and the dominant ST type of each CC is that ST with the highest number of variations in an individual locus [10].

The amplification products obtained were then sequenced according to the method by Sanger at Macrogen Europe, Netherlands.

The nucleotide sequences obtained were analysed using BioNumerics software (version 7.6; Applied Maths, Sint-Martens-Latem, Belgium). Allele and STs were determined using the *Staphylococcus* MLST database at http://saureus.mlst.net/ (accessed on 8 March 2019) and BioNumerics software. The combination of alleles at each locus is comprised of a seven-digit numeric code, which is then compared based on the categorical coefficients and UPGMA (unweighted pair group method with arithmetic mean). 

## 3. Results

### 3.1. Multiplex PCR

Of 237 tested strains, 204 (86.1%) strains contained *spa* fragments, and only 33 (13.9%) did not. Amplified *spa* fragments varied in size, ranging from 180 to 600 bp, depending on the *spa* type. This fragment should amplify in members of *S. aureus* as it is situated within the genome in all members of the species *S. aureus.* Previous studies have reported that the *spa* fragment does not amplify in certain strains of *S. aureus*. Other identification procedures verified that it was indeed *S. aureus*. 

In 10 (4.2%) of the tested strains of *S. aureus* originating from five counties, the gene for methicillin resistance was detected. All strains with detected *mec* gene were resistant toward oxacillin as an indicator of phenotypic resistance toward methicillin (data not presented). The *mecA* and *mecC* genes were each detected in five tested strains of *S. aureus,* while the *PVL* gene was not detected in any of the tested strains. 

Sequencing the *spa* gene successfully types 150 strains (63%). Within the typed samples, 53 different *spa*-types were found, of which four were new to the database. They ranged from 2 (t529) to 12 repetitions (t005, t2678, t774 and NEW 1) in length. The most common *spa*-type was t2678 (14%), followed by t529 (8%), t091, t267 (7.33%), t12029 (6.67%) and t2873 (5.33%). Other *spa* types accounted for less than 5% of the total, with four or fewer isolates for each *spa*-type. 

The distribution of individual *spa* types by county did not illustrate any evident geographic specificities, since most *spa* types were represented with fewer than four specimens of that type per county. The *spa* types that were most common within the tested population were represented in three to seven counties.

Ridom StaphType software was used to analyze the sequencing results, as it directly enables sequence analysis and defines *spa* types with a direct link to the central server. The results are shown in Table 3.

Figure 1 shows the minimum spanning tree (MST) that illustrate the distribution of the *spa* types by county. It is evident that there is no clear geographic association between the individual *spa* types and counties of origin of the samples. Furthermore, certain *spa* types were isolated only in certain counties, though the account for too small a percentage of the tested population to allow for any conclusions. Additionally, the *spa* types that appear at a higher frequency in the population did not show any geographic specificities in their appearance.

### 3.2. Results of Multi-Locus Sequence Typing (MLST)

MLST succeeded in typing 198 of the 237 (83.5%) tested strains. In the typed samples, 32 different allele profiles were defined, including three previously unknown profiles. The most common allele profiles were ST133 (20.7%), ST97 (10.6%), ST352 (10.1%), ST522 (9.1%), ST7 (6.1%), ST504 (5.6%), ST2826 (4.6%), ST20 (4%), ST50 (2.5%), ST1380 (2%), ST6 (2%), ST15 (1.5%), ST101 (1.5%), ST139 (1.5%), ST151 (1.5%), ST398 (1.5%) and ST1094 (1.5%). Other allele profiles (ST) were represented with less than 1% of the total. (Table 4). The presence of allele profiles by county showed no clear geographic specificities (Figure 2). Most allele profiles had fewer than five specimens of the same type and, therefore, no geographic specificities of individual types could be detected. The STs that were most common within the tested population were present in 5–10 counties.

Figure 2 shows the MST typed allele types in comparison to their distribution by county, created using MLST. The figure shows that there is no clear geographic association between the allele types and county of origin. Some allele types were isolated only in certain counties, although their representation in the tested population was too small to allow for any founded conclusions on the association of alleles with geographic origin. Those allele types that appeared at a higher frequency in the population did not show any geographic specificity in their appearance. It should be stressed that MRSA was represented only in three allele profiles: ST130, ST398 and ST97.

Using the eBURST algorithm, six groups (G) and 15 singletons were defined. Group 1 included ST97, ST352, ST2826 and ST71 with type ST97 (CC97) as the dominant type. Group 2 included ST398 and ST3226 (CC398), group 3 ST8 and ST2176 (CC8), group 4 ST7 and ST789 (CC7), and group 5 ST1380 and ST479 (CC479). The final group, group 6 included ST573 and ST1 (CC1). No dominant types were detected in groups 2 to 6.

MLST was the lesser discriminatory technique and resulted in 33 different ST types, also with the highest number of different types represented in the population with a minimum of four samples, even up to nine. This method was also able to type the larger number of samples (83%).

At the national level, strains did not group on a regional basis, although they did group at the international level, with several exceptions, as seen in Figure 3. The MST shows that the STs that are most common in Croatia (e.g., ST 97, 71, 7, 133, 504, 151, 522), also appear in other European countries, such as Germany, France, Netherlands, Switzerland and Ireland.

It is important to note that these conclusions are based on the available data. Most samples in the database are methicillin-resistant strains originating from humans. There are very few strains included that are methicillin-sensitive and originating from animals, as in this study. More research is required on such samples in order to be able to draw conclusions as to the epidemiological importance of certain STs.

## 4. Discussion

Of the 237 tested strains, 204 strains contained a *spa* fragment, while in 33 (13.9%), the *spa* fragment was not proven. These strains were proven in nine counties. In previous studies, *spa* fragments were successfully amplified in 89% of isolates, and a possible reason for non-amplification could be a gene mutation at the site of primer binding [12]. However, other identification procedures confirmed that this was in fact a strain of *S. aureus*.

In our sample, 227 strains (95.8%) were methicillin-sensitive (MSSA) and 10 (4.2%) were MRSA, corroborating previous results. Tianming et al. [29] reported a 93.4% prevalence of methicillin-sensitive *S. aureus* (MSSA), while MRSA was confirmed in 6.6% of isolates in a tested population of dairy cows with mastitis in eastern China. Luini et al. [30] analyzed 160 isolates of *S. aureus* from dairy cows in Italy. They confirmed that 90.8% of isolates were methicillin-sensitive (MSSA), and 9.2% were methicillin-resistant (MRSA). Hamid et al. [31] reported an 83.4% prevalence of MSSA strains and 16.6% of MRSA strains of *S. aureus* isolated from dairy cows in India. Wang et al. [32] examined dairy cow infections with *S. aureus* in six regions of China, and found an 84.1% prevalence of methicillin-sensitive isolates (MSSA), and 15.9% resistant isolates (MRSA). Haran et al. [33] tested 150 pooled milk samples from 50 farms collected over three years, and found 84% of samples contained methicillin-sensitive (MSSA) strains of *S. aureus*.

MRSA was proven in 9.7% of 372 tested herds in Germany [34]. Spohr et al. [35] proved the presence of MRSA on three holdings in Germany, in 5.1 to 16.7% of cows. Locatelli et al. [36] reported the prevalence of MRSA on two farms in Italy, i.e., 3 of 63 (4.8%) tested samples on one farm and 33 of 55 (60%) of tested samples on the second. Basanisi et al. [37] found MRSA in 40 of 484 (8.3%) isolated strains. Vanderhaeghen et al. [8] detected the *mecA* gene in 11 of 118 (9.3%) isolates of *S. aureus* from cows. Pajić et al. [38] reported the presence of *mecA* in 1 of 75 (1.3%) isolates from dairy cow mastitis and in 2 of 11 (18.2%) isolates originating from humans in Serbia, while the *mecC* gene was not detected.

In the present study, the *mecC* gene was proven in 5 (2.1%) out of 237 total *S. aureus*. The *mecC* was also detected in isolates of human and cattle origin in Great Britain and Denmark in 2011. This gene shares only 70% of the nucleotide identity with *mecA.* Garcia Alvarez et al. [39]. Until now, the isolates containing the *mecC* gene have primarily been isolated from cattle, and belonged to the clonal complexes CC130, CC1943 and CC425. Isolates originating from cattle and belonging to CC130 have been described in several European countries (Austria, Denmark, Belgium, France, Germany, Sweden, Netherlands and Great Britain). Humans have also been proven to carry the *mecC* within CC130, in the above countries and also in Slovenia, Switzerland and Spain [38]. Bietrix et al. [40] detected the *mecC* gene in MRSA at four dairy farms, where 22% of cows were positive on one farm and three environmental samples were also positive. All *mecC* positive strains belonged to the CC130 clonal complex. *S. aureus,* carrier of the *mecC* gene has been found in other animal species other than cattle, such as sheep and rabbits, house pets (cats, dogs, hamsters) and wild animals (otters, hedgehogs, wild boar, deer, rats). It has also been proven in storks feeding on remnants of human food. The *mecC* gene in MRSA has also been reported in non-animal sources, such as rivers and city wastewaters in Spain [41,42]. The Panton-Valentine leukocidin (PVL) gene was not proven in any isolates of *S. aureus* in this study. PVL is one of the many toxins produced by *S. aureus*. It is produced by about 2% of strains, and most often appears in skin and soft tissue infections (4.6%), and may cause severe necrotizing pneumonia [39]. Van Duijkeren et al. [43] tested staphylococcus strains isolated from cow mastitis but did not detect the *PVL* gene. Basanisi et al. [37] in isolates originating from dairy cows found the gene that codes for PVL in 50% of isolates. Pajić et al. [38] detected the PVL gene in 5 of 75 (6.7%) from cow mastitis and in 7 of 11 (66.6%) from humans in Serbia.

Considering the large number of *spa* types detected in the present study, there is evidently an exceptional diversity of *spa* types, and no one type was found to be dominant. Nor were there any associations between *spa* types in space (by county) or time (by year). A study conducted in the Netherlands [44] that included strains of methicillin-sensitive *S. aureus* (MSSA) showed partial similarity to this study. Those authors listed t529, t543 and t524 as the most dominant *spa* types. This corroborates our results since the *spa* type t529 was one of the most widely distributed types in our study. Jaki Tkalec et al. [45] detected several *spa* types in Croatia (t005, t011, t091, t073, t164, t4078, t1236, t4460, t015, t527, t728, t9417, t337, t3124, and t5618), though they exclusively tested MRSA strains from cow mastitis cases. In the present study, one MRSA strain belonged to each of the *spa* types t011, t5505 and t843. Spohr et al. [35] proved the presence of *spa* type t011 in dairy cows in Germany. Van Duijkeren et al. [43] found *spa* types t011, t108 and t034 in MRSA isolates in the Netherlands. Vanderhaeghen et al. [8] stated that the most widely distributed *spa* types in Belgium are t011 and t567. Basanisi et al. [37] reported that the most common types in isolates of *S. aureus* in Italy were t355 (67.5% of isolates), t899 and t108 (each 25%) and t127 (5%). In Hungary, Juház-Kaszanyatzky et al. [6] found type t127, while in Denmark, Hasman et al. [28] confirmed the spa types t518, t524 and t529 in cattle. In addition to t529 as the dominant type in Denmark, there were no clear associations between the *spa* types detected in strains of MRSA and MSSA. MSSA shows a lower global diversity of types than MRSA strains, which are more similar at the local level. It is evident that certain *spa* types of MRSA are more common in European populations [46].

Using MLST, we defined 32 different allele profiles and three previously unknown profiles. No association was evident between the presence of allele profiles with the spatial variable (county) since the frequency of appearance of most of the allele profiles was less than five. STs that were most common within the tested population were confirmed in 5–10 counties. Boss et al. [47] used MLST to test isolates of *S. aureus* in mastitis in 12 European countries, and reported that 80% of 281 strains associated with mastitis belonged to only six clonal complexes (CC8, CC97, CC133, CC479 and CC705), while the remaining 20% were *singletons*. The authors stated that CC398 is a clone that is originally of human origin, and that it was transferred to animals, primarily to pigs and cow calves, and that it is a pathogen for dairy cows.

The results presented here correspond to observed trends of allele profile prevalence in strains isolated from animals. Ikawaty et al. [44] did not observe any regional appearance of the individual allele profiles. Of the 85 strains isolated from cases of cattle mastitis, MLST defined the types ST504, ST479, ST71, ST151 and 11 new allele profiles. In 2014–2015, Tianming et al. [29] used MLST to identify 19 allele profiles from 212 strains of *S. aureus* isolated from dairy cow mastitis cases in eastern China. The most prevalent were ST97 (39.2%), ST520 (16%), ST188 (10%), ST398 (7.1%), ST7 (5.2%) and ST9 (4.7%). Rabello et al. [13] stated that CC97 and CC127 were most prevalent in Brazil. In Japan, United Kingdom, United States and Chile, CC97 is dominant [48,49]. Jorgensen et al. [50] listed ST130 (CC3), ST133 and ST132 (CC1) as dominant in Norway. Bergonier et al. [51] examined *S. aureus* of animal origin (LA-MRSA) and found that CC97 and CC133 accounted for 22% of all allele types. They stated that complexes such as CC8, CC22 and CC45 were specific for humans, while CC30 was a transitional form. Voss et al. [52] first proved the presence of ST398 in pigs and farmers in the Netherlands in 2003. Later, the same ST398 was confirmed in Austria, Germany and Denmark, developing into a highly dominant line in Europe and North America [53,54].

From the MLST, it is evident that the STs appearing in Croatia are less common in the rest of Europe and the world. The dendrogram shows that STs such as ST97, ST71, ST7, ST133, ST504, ST151 and ST522, which are most common in Croatia, also appear in other European countries: Germany, France, Netherlands, Switzerland and Ireland. In China, the most prevalent were ST97 (51.9%), ST398 (13.6%) and ST2154 (8.6%) [29] from 2010 to 2013. Half of the researched population consisted of CC97 and CC133 (dominant in animals). CC133 is most often isolated from the milk of small rodents and cows [47,55].

MRSA strains identified during this study belonged to the globally most common MRSA STs: ST398, ST130 and ST97.

In the present study, the allele profiles or STs typical for ruminants were most prevalent (ST133, ST97, ST352, ST522, ST7, ST504). Hasman et al. [28] stated that the most common STs in ruminants are ST71, ST97, ST126, ST133 and ST151, while Smyth et al. [49] listed that the most commonly sequenced types in animals proven by MLST were ST133, ST5, ST71, ST97, ST126 and ST151. Strains of *S. aureus* belonging to ST97 are the most common cause of mastitis in dairy cows in Europe, Asia, and North and South America [48,49].

We used the eBURST algorithm to process the MLST results, which defined six groups and 15 singletons. The most prevalent was group 1 and clonal complex CC97 with 53 strains (26.7%), including ST97, ST352, ST2826 and ST71. These strains are primarily of animal origin, most often cattle, like the strains ST97 and ST352, while isolation from humans indicates their zoonotic transmission [25]. Type ST133 was present in 20.7% and type ST522 in 9.1% of isolates. They are most common in small ruminants, but can also cause mastitis in cattle and oxen [50,56,57]. In this study, CC398 (ST398 and ST3226) was present in 1.5% of isolates. This clone is most common in swine, although its role in causing mastitis in dairy cows and occasionally severe infections in humans has been proven [23,42]. Eight isolates (4%) belonged to the complex CC20. This clone appears in about 1% of cases of human infection, and often causes mastitis in cattle [50].

Although there are differences in the prevalence of different clones on different continents, globally there are only a few clones that are transmitted among humans. The most common in Europe are CC5 (25.7%), CC8 (24.6%) and CC22 (12.1%), and they all fall within the hospital isolates of MRSA (HA-MRSA) [58]. CC8 (ST8 and ST254) and CC22 are often isolated from horses, though they are of human origin and have been proven in equids in Europe [42]. In the present study, two members of the type ST8 (1%) belonging to the clone CC8 and ST22 (singletons) were found. Sakwinska et al. [59] reported CC8 from a case of dairy cow mastitis and also from the farmer. The typical human clone CC8 has been detected in cattle isolates in Belgium [23]. Magro et al. [60] found MRSA CC22 in dairy cows and isolated the agent from the nasal swab of the farmer. The authors stated that CC22 should be considered a potential causative agent of zoonosis, and dairy cows the reservoir.

## 5. Conclusions

Our study is the first research of *S. aureus* from an animal source in the Republic of Croatia that both confirmed and double typed selected strains using molecular methods. Identification of ST-s that is identical with the ST profiles of human origin detected elsewhere implicates a public health threat. The collaboration between public health system and veterinary service is necessary for better insight into epidemiology and public health significance of certain STs, the role of dairy animals in spreading of such staphylococcal strains between bovines and humans as well as the direction of the spread. Furthermore, permanent monitoring of ST-s in humans, bovines and other domestic animals on dairy farms and holdings would enable more efficient preventive measures against the spread of the zoonotic pathogen.

## Figures and Tables

**Figure 1 microorganisms-09-00725-f001:**
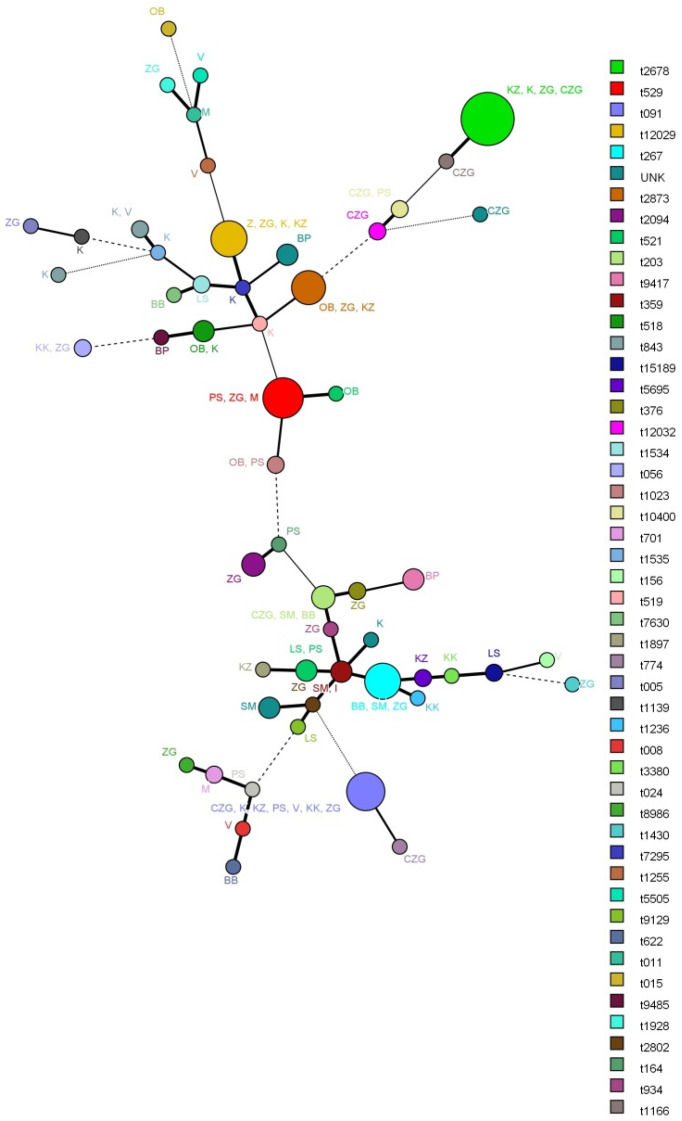
Minimum spanning tree of certain *spa* types according to the county of origin of the samples. This was produced using the *spa clustering* in BioNumerics software (strains are identified according to spy type (different colours according to the legend) and county in Croatia (BB—Bjelovar-Bilogora county; BP—Brod-Posavina county; CZG—City of Zagreb county; I—Istra county; K—Karlovac County; KK—Koprivnica-Križevci county; KZ—Krapina-Zagorje county; LS—Lika-Senj county; M—Međimurje county; OB—Osijek-Baranja county; PS—Požega-Slavonia county; V—Varaždin county; ZG—Zagreb county; UN—unknown). Branch styles correspond to the relationship between strains (thick solid line for differences up to 2.2; thinner solid line for variations up to 3.4; the thinnest solid line for variations up to 4.59; dashed line for differences up to 5.8; and dotted line for variations above 5.8).

**Figure 2 microorganisms-09-00725-f002:**
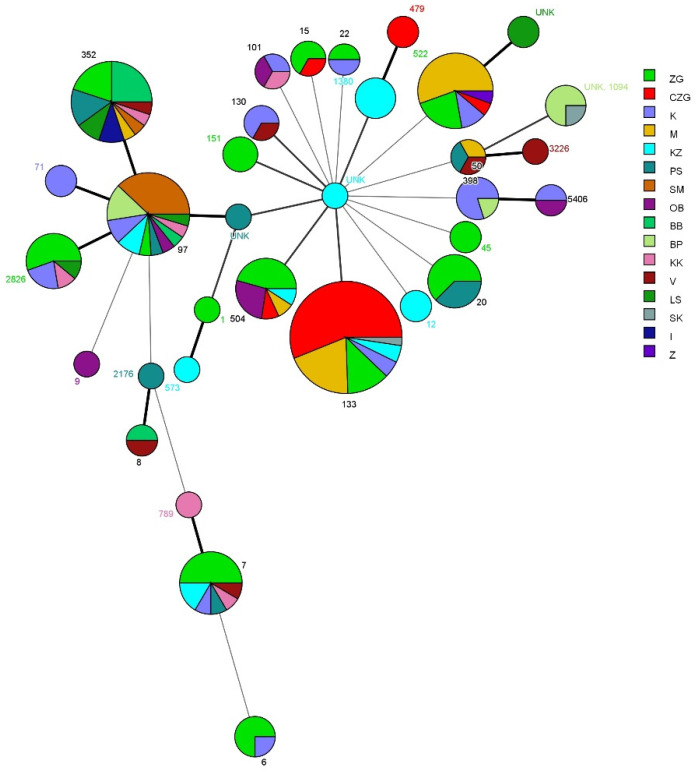
Minimum spanning tree of certain sequence types (ST) according to the county of origin of the samples It was created using BioNumerics software. Colour coding corresponds to different counties in Croatia (BB—Bjelovar-Bilogora county; BP—Brod-Posavina county; CZG—City of Zagreb county; I—Istra county; K—Karlovac County; KK—Koprivnica-Križevci county; KZ—Krapina-Zagorje county; LS—Lika-Senj county; M—Međimurje county; OB—Osijek-Baranja county; PS—Požega-Slavonia county; SK—Šibenik-Knin county; SM—Sisak-Moslavina county; V—Varaždin county; Z—Zadar county; ZG—Zagreb county). Branch styles correspond to the relationship between strains (thick solid line for one locus variants; thinner solid line for double locus variants; the thinnest solid line for three locus variants; dashed line for four locus variants; and dotted line for five locus variants and above).

**Figure 3 microorganisms-09-00725-f003:**
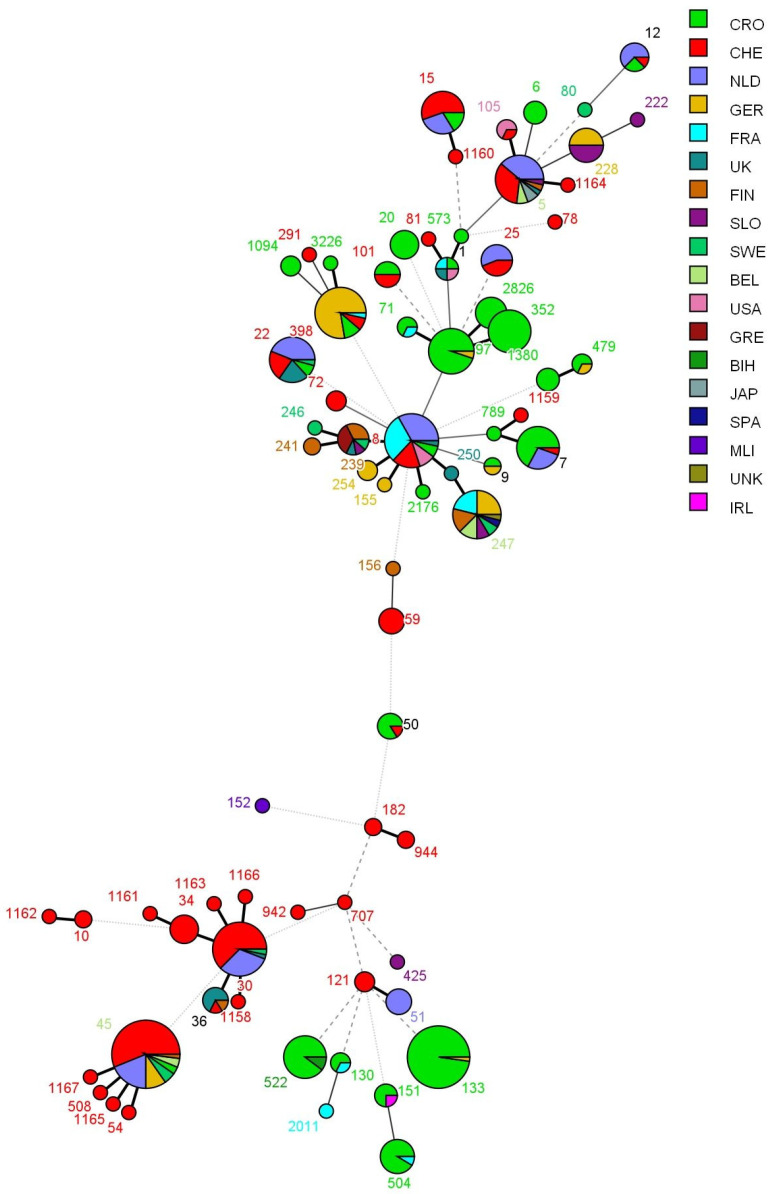
Minimum spanning tree of strains genotyped using the MLST method, from the international database. The strains from the current study are also included. Colour indicates the country of origin, and the number indicates the ST type. Branch styles correspond to the relationship between strains (thick solid line for one locus variants; thinner solid line for double locus variants; the thinnest solid line for three locus variants; dashed line for four locus variants; and dotted line for five locus variants and above).

**Table 1 microorganisms-09-00725-t001:** The number of *S. aureus* isolates included in the research by the year of isolation and the county of origin.

County of	Year					Total
	2014	2015	2016	2017	2018	
Osijek-Baranja	4	3	2	4	2	15
Zagreb	3	37	18	1	-	59
Karlovac	2	1	20	-	1	24
Varaždin	1	-	5	-	1	7
Požega-Slavonija	4	-	9	-	-	13
Međimurje	1	20	2	-	-	23
City of Zagreb	4	3	8	19	-	34
Bjelovar-Bilogora	1	1	5	-	2	9
Koprivnica-Križevci	1	1	7	-	-	9
Krapina-Zagorje	-	2	6	4	4	16
Brod-Posavina	-	-	8	-	-	8
Sisak-Moslavina	-	-	5	4	-	9
Lika-Senj	-	-	3	3	-	6
Šibenik-Knin	-	-	2	-	-	2
Zadar	-	-	-	1	-	1
Istra	-	-	-	-	2	2
Total	21	68	100	36	12	237

**Table 2 microorganisms-09-00725-t002:** Primers used in the research.

Method	Primer	Sequence
*Multiplex PCR*	*spa*-1113F	5′-TAAAGACGATCCTTCGGTGAGC-3′
	*spa*-1514R	5′-CAGCAGTAGTGCCGTTTGCTT-3′
	*mecA* P4	5′-TCCAGATTACAACTTCACCAGG-3′
	*mecA* P7	5′-CCACTTCATATCTTGTAACG-3′
	pvl-F	5′-GCTGGACAAAACTTCTTGGAATAT-3′
	pvl-R	5′-GATAGGACACCAATAAATTCTGGATTG-3′
	*mec_ALGA251_* MultiFP	5′-GAAAAAAAGGCTTAGAACGCCTC-3′
	*mec_ALGA251_* MultiRP	5′-GAAGATCTTTTCCGTTTTCAGC-3′
Spa typing	spaAF1	5′-GACGATCCTTCGGTGAGC-3′
	spaAR1	5′-CAGCAGTAGTGCCGTTTGC-3′
MLST	arc up	5′-TTGATTCACCAGCGCGTATTGTC-3′
	arc dn	5′-AGGTATCTGCTTCAATCAGCG-3′
	aro up	5′-ATCGGAAATCCTATTTCACATTC-3′
	aro dn	5′-GGTGTTGTATTAATAACGATATC-3′
	glp up	5′-CTAGGAACTGCAATCTTAATCC-3′
	glp dn	5′-TGGTAAAATCGCATGTCCAATTC-3′
	gmk up	5′-ATCGTTTTATCGGGACCATC-3′
	gmk dn	5′-TCATTAACTACAACGTAATCGTA-3′
	pta up	5′-GTTAAAATCGTATTACCTGAAGG-3′
	pta dn	5′-GACCCTTTTGTTGAAAAGCTTAA-3′
	tpi up	5′-TCGTTCATTCTGAACGTCGTGAA-3′
	tpi dn	5′-TTTGCACCTTCTAACAATTGTAC-3′
	yqi up	5′-CAGCATACAGGACACCTATTGGC-3′
	yqi dn	5′-CGTTGAGGAATCGATACTGGAAC-3′

**Table 3 microorganisms-09-00725-t003:** Results of typing the tested strains of *S. aureus* by amplifying and sequencing the *spa* fragments, overview of identified *spa*-types and appearance of repeats specific for an individual type.

*spa*-Type	N Strains	%	N Counties	MRSA	MSSA	*spa*-Type Repeats
NEW 1	1	0.67	1	-	1	07-23-12-21-22-17-34-34-34-34-33-34
NEW 2	1	0.67	1	-	1	08-25-51-68-02-24-02-24
NEW 3	3	2.00	1	-	3	08-16-16-17
NEW 4	3	2.00	1	-	3	07-23-21-17-13-34-33-34
t005	1	0.67	1	-	1	26-23-13-23-31-05-17-25-17-25-16-28
t008	1	0.67	1	-	1	11-19-12-21-17-34-24-34-22-25
t011	1	0.67	1	1	-	08-16-02-25-34-24-25
t015	1	0.67	1	-	1	08-16-02-16-34-13-17-34-16-34
t024	1	0.67	1	-	1	11-12-21-17-34-24-34-22-25
t056	2	1.33	2	-	2	04-20-12-17-20-17-12-17-17
t091	11	7.33	7	-	11	07-23-21-17-34-12-23-02-12-23
t1023	2	1.33	2	-	1	07-06-34 *
t10400	2	1.33	2	-	2	03-16-16-17-17-17-17-23-24
t1139	1	0.67	1	-	1	26-23-13-17-25-17-25-16-28
t1166	1	0.67	1	-	1	03-16-21-17-23-13-17-17-17-23-24
t12029	10	6.67	4	-	10	04-31-17-24-17
t12032	2	1.33	1	-	2	03-16-16-17-17-17-23-24
t1236	1	0.67	1	-	1	26-23-12-21-17-34-34-34-33-34
t1255	1	0.67	1	-	1	08-16-34-24-25
t1430	1	0.67	1	-	1	07-16-23-02-12-23-02-34
t15189	2	1.33	1	-	2	07-23-12-21-33-34
t1534	2	1.33	1	-	2	04-31-17-25-17-17
t1535	1	0.67	1	-	1	04-82-17-25-17-25-16-17
t156	1	0.67	1	-	1	07-23-12-33-22-17
t164	1	0.67	1	-	1	07-06-17-21-34-34-22-34
t1897	1	0.67	1	-	1	26-23-34-21-17-34-34-34-34-33-34
t1928	1	0.67	1	-	1	08-02-25-02-25-34-24-25
t203	4	2,67	3	-	4	07-23-12-12-34-34-33-34
t2094	4	2.67	1	-	4	26-06-17-21-34-34-22-34
t267	11	7.33	3	-	11	07-23-12-21-17-34-34-34-33-34
t2678	21	14.00	4	-	21	03-16-12-21-17-23-13-17-17-17-23-24
t2802	1	0.67	1		1	07-23-21-17-34-34-34-33-34
t2873	8	5.33	3	-	8	04-20-17-31-24
t3380	1	0.67	1	-	1	07-23-12-21-17-34-34
t359	3	2.00	2	-	3	07-23-12-21-17-34-34-33-34
t376	2	1.33	1	-	2	07-23-12-34-34-34-33-34
t518	3	2.00	2	-	3	04-20-17-23-24-20-17-25
t519	1	0.67	1	-	1	04-20-17-25
t521	4	2.67	3	-	4	07-23-12-21-17-34-34-34-34-33-34
t529	12	8.00	3	1	11	04-34
t5505	1	0.67	1	1		08-16-315-25-02-25-34-24-25
t5695	2	1.33	1	-	2	07-23-12-21-17-34-34-34-34-34
t622	1	0.67	1	-	1	11-19-12-21-17-34-22-25
t701	2	1.33	1	-	2	11-10-21-17-34-24-34-22-25-25
t7295	1	0.67	1	-	1	04-31-17-17
t7630	1	0.67	1	-	1	04-31-17-25-17-17-17-17
t774	1	0.67	1	-	1	07-23-12-34-34-12-12-12-23-02-12-23
t843	3	2.00	2	1	2	04-82-17-25-17-25-25-16-17
t8986	1	0.67	1	-	1	11-10-21-17-34-24-34-17-25-25
t9129	1	0.67	1	-	1	07-16-21-17-34-34-34-33-34
t934	1	0.67	1	-	1	07-23-12-34-34-34-34-33-34
t9417	3	2.00	1	-	3	15-17-34-34-34-34-33-34
t9485	1	0.67	1	-	1	04-20-23-24-20-17-25

* one strain with the *spa* gene undetected with the Multiplex polymerase chain reaction (PCR); N strain—the number of strains belonging to a spa type; N counties—the number of counties in which a spa type has been detected.

**Table 4 microorganisms-09-00725-t004:** Identified multi-locus sequence typing (MLST) gene sequences allele profiles per number of isolates and eBURST (Based Upon Related Sequence Types) classification.

ST	N Isolates *	ST (%)	Alellic Profile							eBURST
			*arcC*	*aroE*	*glpF*	*gmk*	*pta*	*tpi*	*yqiL*	
1	1	0.51	1	1	1	1	1	1	1	G6
6	4	2.02	12	4	1	4	12	1	3	single
7	12	6.06	5	4	1	4	4	6	3	G4
8	2	1.01	3	3	1	1	4	4	3	G3
9	1	0.51	3	3	1	1	1	1	10	single
12	2	1.01	1	3	1	8	11	5	11	single
15	3	1.52	13	13	1	1	12	11	13	single
20	8	4.04	4	9	1	8	1	10	8	single
22	2	1.01	7	6	1	5	8	8	6	single
45	2	1.01	10	14	8	6	10	3	2	single
50	5	2.53	16	16	12	2	13	13	15	G7
71	2	1.01	18	1	1	1	1	5	3	G1
97	21 (4)	10.61	3	1	1	1	1	5	3	G1
101	3	1.52	3	1	14	15	11	19	3	single
130	3 (3)	1.52	6	57	45	2	7	58	52	single
133	41	20.71	6	66	46	2	7	50	18	single
151	3	1.52	6	72	12	43	49	67	59	single
352	20	10.10	3	78	1	1	1	5	3	G1
398	3 (3)	1.52	3	35	19	2	20	26	39	G2
479	2	1.01	52	79	54	18	56	32	65	G5
504	11	5.56	6	72	72	43	52	67	59	single
522	18	9.09	18	95	45	2	7	15	5	single
573	1	0.51	1	1	1	1	12	1	1	G6
789	1	0.51	3	4	1	4	4	6	3	G4
1094	3	1.52	10	35	19	2	49	26	39	single
1380	4	2.02	169	79	54	18	56	32	65	G5
2176	1	0.51	3	3	1	1	4	243	3	G3
2826	10	4.55	3	1	1	1	309	5	3	G1
3226	1	0.51	3	35	19	2	20	352	39	G2
5858	1	0.51	52	79	54	18	56	32	18	G5
5859	1	0.51	3	78	1	1	1	45	3	G1
5406	2	1.01	3	16	12	2	13	13	15	G7

* Values in brackets represent number of MRSA: ST—sequence type; N isolates—the number of strains belonging to a ST; ST (%)—percent of ST among successfully typed strains.

## Data Availability

The datasets generated and analyzed during the current study are available from the corresponding author upon reasonable request.

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
