# Peer review of "Multi Locus Sequence Typing and spa Typing of Staphylococcus aureus Isolated from the Milk of Cows with Subclinical Mastitis in Croatia"

_microorganisms, 2021, doi:10.3390/microorganisms9040725_

Round 1

Reviewer 1 Report

The aim of this study was to provide a molecular epidemiological profile of two hundred and thirty-seven S. aureus isolates from the milk of cows with subclinical mastitis using MLST, spa Typing, and to investigate the presence of the mecA, mecC, and pvl genes. The study provides potentially valuable data on the contribution of different spa types and MLST to the subclinical mastitis cases in dairy cows in Croatia. Overall, this study contributes to the knowledge gap regarding Staphylococcus strains in different regions of the world. In my opinion, the paper is suitable for publication in the journal after very minor corrections.

Lines 47-50: This paragraph should be rewritten to refer to the specific use of MLST in this species.

Line 70: The definition of the LA -MRSA abbreviation is not included.

Lines 92: It is presumed that "samples" are milk samples, but this is not stated anywhere in the section.

Lines 94-96: For using Micronaut RPO, blood agar and not Baird-Parker agar should be used. In addition, reference information for Aesculin Blood agar and Baird Parker agar should be included. Gram and catalase tests must be done beforehand, not just the coagulase test.

Line 110/111 The sentence should be reworded to: "To obtain DNA for multiplex PCR" ...

Line 136: The authors imply that "PCR replicates the spa region, sequencing it..." which is incorrect. The sentence should be rewritten because PCR is not a sequencing technique. After PCR, the resulting fragments should be sequenced for spa typing.

Line 134: this is table 1

Line 159: this is table 2; abbreviations should be written in full in the legend (for example ST)

Line 191: this is table 3; abbreviations should be written in full in the legend

Lines 194 & 208: inpossible to read these figures

Author Response

Reviewer 1

Comment

Action

Revised version

Lines 47-50: This paragraph should be rewritten to refer to the specific use of MLST in this species.

The text has been rephrased

The Multi Locus Sequence Typing (MLST) of S. aureus is based on the analysis of seven house keeping genes (arcC, aroE, glpF, gmk, pta, tpi and ygiL) according to Enright et al. [10]. Order of nucleotides within genes enables to differentiate strains, while the combination of alleles make the allele profile or Sequence type (ST) [11].

Line 70 The definition of the LA -MRSA abbreviation is not included.

the definition of LA-MRSA is included

Livestock-associated meticillin resistant S. aureus (LA-MRSA)

LINE 92 It is presumed that "samples" are milk samples, but this is not stated anywhere in the section.

„milk“ is included

…quarter udder milk samples…

Line 94/96 For using Micronaut RPO, blood agar and not Baird-Parker agar should be used. In addition, reference information for Aesculin Blood agar and Baird Parker agar should be included. Gram and catalase tests must be done beforehand, not just the coagulase test.

The text related on the identification has been rephrased; reference is added

Suspected colonies were Gram stained, checked for catalase production and subcultivated on Baird-Parker agar and blood agar [25]. Furthermore the tube coagulase-test was carried out as well as the identification by semi automated identification system Micronaut RPO (Merlin Diagnostika GmbH, Germany).

National Mastitis Council. Staphylococci. In Laboratory Handbook on Bovine Mastitis, 3rd ed.; National Mastitis Council: Verona, WI, USA, 2017. 

Line 110/111 The sentence should be reworded to: "To obtain DNA for multiplex PCR" ...

The sentence has been reworded

In order to obtain DNA isolation from strains for further genotyping was performed using the commercially available kit QIAcube DNA Mini Kit (QIAGEN, Hilden, Germany) was used according to the manufacturer’s instructions.

Line 136 The authors imply that "PCR replicates the spa region, sequencing it..." which is incorrect. The sentence should be rewritten because PCR is not a sequencing technique. After PCR, the resulting fragments should be sequenced for spa typing.

The sentence has been rephrased

The fragments obtained by amplification of the spa region by the polymerase chain reaction (PCR) were sequenced and assigned by numerical code that is based on the order of nucleotides.

Line 134: this is table 1

Corrected

Line 159: this is table 2; abbreviations should be written in full in the legend (for example ST)

Corrected

Line 191: this is table 3; abbreviations should be written in full in the legend

Corrected

Lines 194 & 208: impossible to read these figures

Corrected

Reviewer 2 Report

This manuscript presents the analysis by spa typing and MLST of 237 S. aureus isolated from dairy cows in Croatia. The data presented are interesting but some modifications are needed for this manuscript to be published, in my opinion.

Please have your manuscript proof read to improve the quality of the English language.

Tables are supposed to be numbered in their order of appearance in the text. Please do so in your manuscript.

Please check for the homogeneity of the police size (L54-60 and elsewhere)

There is a discrepancy between the detection of mec genes and the resistance to methicillin or analogues. Were the strains identified as MRSA by PCR validated by antimicrobial testing ?

It is possible to submit new spa types to the data repository. Please do so and then used the new attributed spa types in your manuscript.

The quality of the figures must be improved to be readable.

Please add a table describing the strains used in this study. There number according to the county and the year of isolation.

Specific comments :

Introduction :

Please provided more information about the situation in Croatia : what is the prevalence of S. aureus in mastitis.

Methods :

L91: Please italicize S. aureus

L92 : Please define your criteria for subclinical mastitis.

L46 : I suggest to remove this sentence.

L70 : please define LA-MRSA the first time you use it.

L82 : please replace by MSSA that was defined L64.

L126 : replace replication by amplification

L127 : spa gene

L150 : the MLST genes are not shown in table 2 that should be table 1.

Results:

L200: Remove the xc after ST352

Discussion:

Please review the paragraph organization of the discussion. The paragraph L261-265 could be merged with the precedent one. Same for the 2 paragraphs L266-279. Please Make sure that each paragraph discusses one point and that the point is not repeated later on.

L245-247: When PCR is used the verb amplify is better than replicate.

L266-267: It seems that this study is focusing on dairy milk. Enough examples are provided, I would suggest to remove that

L270: Please clarify what you mean by “any research”

L271: Please revise the percentage. It is 5 out of 237 total S. aureus (2.1%) or 5 out of the 10 MRSA (50 %).

L273: Please add a reference.

L280-287: This should be merged with the precedent paragraph.

L303 : please precise the origin of these strains (human, dairy ?)

L306: correct andi

L295-316: In this paragraph please focus only on spa types and precise the origin of the strains.(PLV gene is discussed earlier in the discussion, please merge the information.

Tables and figures:

All tables and figures should be self-explanatory. Please provide more information for each of them

Table 2 : MLST gene sequences are missing

Table 3 : Is it really MRSA ? Not S. aureus ? Are the data presented here your own results ? If yes, they should be cited in the result section not the material one.

Table 1: Please be more precise in your description “one strain without the spa gene”. How a strain without the spa gene can have a spa type ? You mean that for this strain the spa gene was not detected in the multiplex PCR using different primers.

I would suggest to classify the spa type according to their abundance (%) rather than their number.

Figures 1 and 2: The resolution of the figures must be improved to be readable. Please explain the legend (county) and define the abbreviations used in the legend. Can you replace “key” by the number of the strain if that is what it is?

Figure 3: The resolution of the figures must be improved to be readable. Please explain the legend (country) and define the abbreviations used in the legend.

Author Response

Reviewer 2

Comment

Action

Added statements

Please have your manuscript proof read to improve the quality of the English language.

The manuscript has been proofed by genuine English user. The certificate is attached.

Tables are supposed to be numbered in their order of appearance in the text. Please do so in your manuscript.

The table numbering has been corrected.

Please check for the homogeneity of the police size (L54-60 and elsewhere)

The font size has been corrected.

There is a discrepancy between the detection of mec genes and the resistance to methicillin or analogues. Were the strains identified as MRSA by PCR validated by antimicrobial testing ?

The resistance pattern was evaluated for 13 routinely used antimicrobials and oxacillin in all 237 strains by disc-diffusion method. All mec bearing strains strains were resistant toward oxacillin. However these data are not presented. A statement on the resistance toward oxacillin has been included in the results section of the manuscript.

All strains with detected mec gene were resistant toward oxacillin as an indicator of phenotypic resistance toward methicillin (data not presented).

It is possible to submit new spa types to the data repository. Please do so and then used the new attributed spa types in your manuscript.

We are aware that presentation of the new spa types would be the best way of presentation. Unfortunately the assigning of a new designation depends upon an administrator in the repository. We have repeatedly claimed for new spa designations recently. At the moment we can only try to convince the reviewer by writing the application number for our strains that we sent to repository since we have been still waiting for assigning the new spa types.  Our application/claims are listed bellow

T21032202 (strain CVI_9);

T21032203 (strain CVI_15);

T21032204 (strain CVI_100);

T21032205 (strain CVI_161).

The quality of the figures must be improved to be readable.

In the original manuscript we included dendrograms. Since it is technically impossible to make the dendrograms to be large enough and have acceptable resolution at the same time, in the revised manuscript we presented results of spa typing and MLST as Minimum spanning tree. The figures have resolution of 600 dpi. 

Please add a table describing the strains used in this study. There number according to the county and the year of isolation.

New table has been added.

Please provided more information about the situation in Croatia : what is the prevalence of S. aureus in mastitis.

Additional text has been added.

Cvetnić et al. (2016) investigated quarter udder milk samples from 385 cows in Croatian dairy farms and found 13.1% infected udder quarters. At least one udder quarter was infected in 29.2% of cows examined.  S. aureus was isolated from 4.48% of udder quarter samples.

L91: Please italicize S. aureus

Has been done

L92 : Please define your criteria for subclinical mastitis.

Criteria has been

Cows were considered as having subclinical mastitis based on positive California mastitis test (CMT) and pathogen isolation without clinical signs of mastitis.

L46 : I suggest to remove this sentence.

The sentence has been removed.

L70 : please define LA-MRSA the first time you use it.

LA-MRSA has been defined

L82 : please replace by MSSA that was defined L64.

It has been corrected

L126 : replace replication by amplification

„replication“ has been changed with „amplification“

L127 : spa gene

Has been corrected

L150 : the MLST genes are not shown in table 2 that should be table 1.

Table numbering is corrected as well as numbering in the text.

L200: Remove the xc after ST352

Has been erased

Please review the paragraph organization of the discussion. The paragraph L261-265 could be merged with the precedent one. Same for the 2 paragraphs L266-279. Please Make sure that each paragraph discusses one point and that the point is not repeated later on.

The paragraphs have been merged into one.

L245-247: When PCR is used the verb amplify is better than replicate.

„replicated“ has been changed with „amplified“

L266-267: It seems that this study is focusing on dairy milk. Enough examples are provided, I would suggest to remove that

L270: Please clarify what you mean by “any research”

The sentence has been shortened.

Pajić et al. [38] reported the presence of mecA in 1 of 75 (1.3%) isolates from dairy cow mastitis and in 2 of 11 (18.2%) isolates originating from humans in Serbia, while the mecC gene was not detected.

L271: Please revise the percentage. It is 5 out of 237 total S. aureus (2.1%) or 5 out of the 10 MRSA (50 %).

The statement has been corrected.

In the present study, the mecC gene was proven in 5 (2.1%) out of 237 total S. aureus.

L273: Please add a reference.

Reference has been added.

García-Álvarez, L.; Holden, M.T.; Lindsay, H.; Webb, C.R.; Brown, D.F., Curran, M.D.; Walpole, E.; Brooks, K.; Pickard, D.J.; Teale, C.; Parkhill, J.; Bentley, S.D.; Edwards, G.F.; Girvan, E.K.; Kearns, A.M.; Pichon, B.; Hill, R.L.; Larsen, A.R.; Skov, R.L.; Peacock, S.J.; Maskell, D.J.; Holmes, M.A. Meticillin-resistant Staphylococcus aureus with a novel mecA homologue in human and bovine populations in the UK and Denmark: a descriptive study. Lancet Infect Dis. 2011 Aug;11(8):595-603. doi: 10.1016/S1473-3099(11)70126-8. PMID: 21641281; PMCID: PMC3829197.

L280-287: This should be merged with the precedent paragraph.

The paragraph has been merged with the previous one.

L303 : please precise the origin of these strains (human, dairy ?)

The statement has been filled up.

Jaki Tkalec et al. [45] detected several spa types in Croatia (t005, t011, t091, t073, t164, t4078, t1236, t4460, t015, t527, t728, t9417, t337, t3124, and t5618), though they exclusively tested MRSA strains from cow mastitis cases.

L306: correct andi

Has been corrected.

L295-316: In this paragraph please focus only on spa types and precise the origin of the strains.(PLV gene is discussed earlier in the discussion, please merge the information.

The statements have been revised; facts regarding the PVL mentioned earlier in the text have been omitted.

Tables and figures:

All tables and figures should be self-explanatory. Please provide more information for each of them

The resolution of the figures has been improved, tables have been corrected according to the specific comments.

Table 2 : MLST gene sequences are missing

The Table heading has been corrected.

Table 3 : Is it really MRSA ? Not S. aureus ? Are the data presented here your own results ? If yes, they should be cited in the result section not the material one.

The column heading has been changed. An asterisk with the explanation has been written bellow the table.

Table 1: Please be more precise in your description “one strain without the spa gene”. How a strain without the spa gene can have a spa type ? You mean that for this strain the spa gene was not detected in the multiplex PCR using different primers

The text has been changed according the reviewer's suggestion.

I would suggest to classify the spa type according to their abundance (%) rather than their number.

Authors agree with the reviewer regarding the way of data presentation. Hence both, absolute numbers (N) as well as percent of each spa-type have been included in the table. 

Figures 1 and 2: The resolution of the figures must be improved to be readable. Please explain the legend (county) and define the abbreviations used in the legend. Can you replace “key” by the number of the strain if that is what it is?

Corrected

Figure 3: The resolution of the figures must be improved to be readable. Please explain the legend (country) and define the abbreviations used in the legend.

Corrected

Table 1. The number of S. aureus isolates included in the research by the year of isolation and the county of origin

County of

Year 

Total 

2014

2015

2016

2017

2018

Osijek-Baranja

4

3

2

4

2

15

Zagreb

3

37

18

1

-

59

Karlovac

2

1

20

-

1

24

Varaždin

1

-

5

-

1

7

Požega-Slavonija

4

-

9

-

-

13

Međimurje

1

20

2

-

-

23

City of Zagreb

4

3

8

19

-

34

Bjelovar - Bilogora

1

1

5

-

2

9

Koprivnica - Križevci

1

1

7

-

-

9

Krapina - Zagorje

-

2

6

4

4

16

Brod - Posavina

-

-

8

-

-

8

Sisak - Moslavina

-

-

5

4

-

9

Lika - Senj

-

-

3

3

-

6

Šibenik - Knin

-

-

2

-

-

2

Zadar

-

-

-

1

-

1

Istra

-

-

-

-

2

2

Total

21

68

100

36

12

237

Reviewer 3 Report

The manuscript of MarkoSamardžija represents the first research of S. aureus from an animal source in  Croatia that both confirmed and double typed selected strains using molecular methods. Some of the analysed strains belonged to the same MLST profiles as isolates of human origin detected in other studies implicating possible zoonotic nature. This paper represents a clear milestone of how One Health research should be conducted in every country, because animals (especially food animals) and human share the same pathogens and it is important to monitor circulating AMR. The work is clearly presented and could be accepted in microorganisms journal also if the technologies applied are not exactly the latest. But this work is important to fix the point in circulating AMR.

Minor concerns/ integration:

Introduction:

lane 54- please correct the font used

lane 80: please integrate the introduction with findinngs/ perspectives of these articles: doi: 10.3390/ani10122378. and doi: 10.3390/ijms21061914.

Figure 1, 2 and 3 are of poor quality and the resolution must be improved.

Conclusions: Please authors make consideration of One Health Approach. See for examples doi: 10.2147/IDR.S272733. and doi:10.1039/c5mb00788g.

Author Response

Reviewer 3

Comment relying on

Action

Revised version

Line 54 please correct the font used

The font has been corrected

Line 80 please integrate the introduction with findings/ perspectives of these articles: doi: 10.3390/ani10122378. and doi: 10.3390/ijms21061914.

New reference has been included

Figure 1, 2 and 3 are of poor quality and the resolution must be improved.

Corrected

Conclusions: Please authors make consideration of One Health Approach. See for examples doi: 10.2147/IDR.S272733. and doi:10.1039/c5mb00788g.

Our study is the first research of S. aureus from an animal source in the Republic of Croatia that both confirmed and double typed selected strains using molecular methods. Identification of ST-s that is identical with the ST profiles of human origin detected elsewhere implicates a public health thread. The collaboration between public health system and veterinary service is necessary for better insight into epidemiology and public health significance of certain ST-s, the role of dairy animals in spreading of such staphylococcal strains between bovines and humans as well as the direction of the spread. Furthermore permanent monitoring of ST-s in humans, bovines and other domestic animals on dairy farms and holdings would enable more efficient preventive measures against the spread of the zoonotic pathogen.

Round 2

Reviewer 2 Report

Thank you for the revision of the manuscript and your answers to my comments.

3 more points:

-L70 : Please add (LA-MRSA)

-Table 2 : MLST gene primers are missing

-L426 : Do you really mean thread or rather threat ?

Author Response

Dear reviewer. Thank you!

-L70 : Please add (LA-MRSA)

addeed

-Table 2 : MLST gene primers are missing

MLST gene primers added in the Table 2.

-L426 : Do you really mean thread or rather threat ?

Sorry, of course we mean threat. Corrected.